



**Options for mitigating global warming potential of a double-rice field in China**
Guangbin Zhang[1], Haiyang Yu[1,2], Xianfang Fan[1,2], Yuting Yang[1,2], Jing Ma[1], and Hua
Xu[1]
[1] State Key Laboratory of Soil and Sustainable Agriculture, Institute of Soil Science, Chinese Academy
of Sciences, Nanjing 210008, China
[2] University of Chinese Academy of Sciences, Beijing 100049, China
*Correspondence to*: Hua Xu (hxu@issas.ac.cn)
**Abstract.** Traditional land managements (neither drainage nor tillage, NTND) in winter fallow season
result in substantial $CH_4$ and $N_2O$ emissions from the double-rice fields in China. For investigating the
effects of drainage and tillage in winter fallow season on global warming potentials (GWPs) of $CH_4$ and
$N_2O$ emissions and developing mitigation options, a field experiment with four treatments: NTND,
drainage but non-tillage (NTD), tillage but non-drainage (TND), and both drainage and tillage (TD) were
carried out from 2010 to 2014 in a Chinese double-rice field. In winter fallow season total precipitation
and mean daily temperature had important effects on $CH_4$ emission, and significant correlations were
observed between them and $CH_4$ emission. Compared with NTND, drainage and tillage reduced $CH_4$
emission in early- and late-rice seasons and decreased annual emission by 54 and 33 kg $CH_4$ ha$^{-1}$ yr$^{-1}$,
respectively. Drainage and tillage increased $N_2O$ emission in winter fallow season while reduced it in
early- and late-rice seasons, causing annual $N_2O$ emission unaffected. Accordingly, the GWPs were
decreased by 1.49 and 0.92 t $CO_2$-eq ha$^{-1}$ yr$^{-1}$, respectively, and they were far more reduced by
combining drainage with tillage, with a mitigation potential of 1.96 t $CO_2$-eq ha$^{-1}$ yr$^{-1}$. Low total C
content and high C/N ratio in rice residues revealed that tillage in winter fallow season reduced $CH_4$ and
$N_2O$ emissions in early- and late-rice seasons. Moreover, drainage and tillage significantly decreased the
abundance of methanogens in paddy soil, which was a possible reason for the decrease of $CH_4$ emission.
Greenhouse gas intensity was significantly decreased by drainage and tillage, and it was much more
reduced by combining drainage with tillage, with a reduction of 0.17 t $CO_2$-eq t$^{-1}$ yield yr$^{-1}$. The results
indicate that soil drainage combined with tillage in winter fallow season is an effective mitigating
strategy in double-rice fields.




## 1 Introduction

Methane ($CH_4$) and nitrous oxide ($N_2O$) are two of the most important greenhouse gases (GHGs) after carbon dioxide ($CO_2$) in the atmosphere. According to the Greenhouse Gas Bulletin of World Meteorological Organization, the concentrations of atmospheric $CH_4$ and $N_2O$ reached at 1824 and 325.9 ppb in 2013, respectively (WMO, 2014). Paddy fields are considered to be the major sources of atmospheric $CH_4$ and $N_2O$. Since the 2000s, effective options for mitigating $CH_4$ and $N_2O$ emissions from paddy fields have been continually explored over the world (McCarl and Schneider, 2001; Yan et al., 2005; Hussain et al., 2015), i.e. modifying irrigation and fertilization patterns (Cai et al., 2003; Hussain et al., 2015; Linquist et al., 2015), setting integrated soil–crop system management practices (Chen et al., 2014; Zhang et al., 2013b), and selection of suitable rice cultivar with high production but low GHGs emissions (Su et al., 2015; Hussain et al., 2015; Ma et al., 2010b), etc. Nevertheless, potential mitigating methods might be still available due to the diversity of rice-based ecosystems and the difference in agronomic management practices (Weller et al., 2016).

China is one of the largest rice producers in the world, and its harvested area contributes 18.5% of the world total (FAOSTAT, 2013). In China, total $CH_4$ and $N_2O$ emissions from paddy fields were estimated to be 6.4 Tg $yr^{-1}$ and 180 Gg $yr^{-1}$, respectively (Zhang et al., 2014). Double rice is the major rice-cropping system in China, accounting for over 40% of total rice cultivation area (Yearbook, 2013) and emitting ca. 50% of the total paddy $CH_4$ in China (Zhang et al., 2011b; Chen et al., 2013). Double-rice fields mainly distribute at the south of the Yangtze River where usually has relative large precipitation and high temperature in winter fallow season. Traditionally, the fields are fallow in winter season with the soil neither drainage nor tillage after late-rice harvest, and they are usually subjected to visible floodwater after a heavy or a long-time raining. It is very likely to bring about $CH_4$ emission from these fields in winter fallow season and further to promote its emission during the following rice growth season. Modeling data had shown that $CH_4$ emission was significantly correlated with simulated soil moisture and mean precipitation of the preceding non-rice growth season (Kang et al., 2002). Incubation and pot experiments also affirmed that the higher the soil water contents in the non-rice growth season, the higher the $CH_4$ production rates and the more the $CH_4$ emissions in the subsequent rice season (Xu et al., 2003). An available mitigating option is hence proposed in this region, that is, the fields are drained to decrease the accumulation of rainwater in winter fallow season and finally to attenuate the positive effect of winter precipitation on $CH_4$ emission. However, drainage possibly stimulates $N_2O$ emission



from paddy field in winter fallow season because soil water content changes more quickly and
intensively. It is well recognized that soil moisture regulates the processes of denitrification and
nitrification and thus $N_2O$ emission (Lan et al., 2013; Bateman and Baggs, 2005). Since the overall
balance between the net exchange of $CH_4$ and $N_2O$ emissions constitutes global warming potentials
(GWPs) of rice ecosystem, the effect of soil drainage in winter fallow season on mitigating GWPs
year-round from the double-rice field is not well understood.

67        Soil tillage is a conventional practice in rice cultivation, and considerable reports have shown that

tilling the soil prior to rice transplanting plays a key role in $CH_4$ and $N_2O$ emissions (Hussain et al., 2015;
Zhao et al., 2016). Meanwhile, tillage after rice harvest in winter fallow season probably has very
important effects on $CH_4$ and $N_2O$ emissions. Firstly, it is beneficial for the rainwater to penetrate into
the subsoil, which won't lead to the accumulation of rainwater in winter fallow season. It is then difficult
to form the strict anaerobic environments in the top soil, which not only reduces $CH_4$ emission directly
during the non-rice growing season, but also indirectly inhibits $CH_4$ emission during the following rice
season. On the contrary, tillage makes rice residues fully contact with the soil and microorganism, which
may accelerate the decomposition of organic matters and then in favor of $CH_4$ production and emission
in the non-rice growth season (Pandey et al., 2012; Hussain et al., 2015). Secondly, it may also play a
key role in $CH_4$ emission during the following rice season owing to the incompletely decomposed rice
residues. In addition, tillage in winter fallow season whether increases $N_2O$ emission from the field or
not is still not very clear. There are some contradictive lines of evidence asserting the promotion and
reduction in $N_2O$ emissions from rice fields by soil tillage. For instance, tillage changes the soil
properties (soil porosity and soil moisture, etc.) and then promotes $N_2O$ emission (Pandey et al., 2012;
Mutegi et al., 2010) whereas incorporation of rice residues due to tillage may reduce $N_2O$ emission as a
result of N immobilization (Huang et al., 2004; Ma et al., 2010a). Based on a 3-year field measurement
(Shang et al., 2011), the possible agricultural mitigating strategy that is crop residues incorporated into
the soil accompanying with drainage in winter fallow season, has been proposed in a double-rice field.
Nevertheless, the effects of drainage combined with tillage in winter fallow season on annual $CH_4$ and
$N_2O$ emissions from double-rice fields, in particular on the corresponding mitigation potential are
scarcely documented.

89        An *in situ* field measurement was conducted year-round for 4 years from 2010 to 2014 to study the

$CH_4$ and $N_2O$ emissions from a typical double-rice field in China. The objectives of this study are (1) to





investigate the effects of soil drainage and tillage in winter fallow season on $CH_4$ and $N_2O$ emissions
from the paddy field, (2) to estimate the mitigation potential of drainage and tillage, and thereby (3) to
suggest the optimal land management strategies in winter fallow season for reducing GWPs of $CH_4$ and
$N_2O$ emissions in the double rice-cropping systems in China.

**2 Methods and materials**
**2.1 Field site and experimental design**
The experimental field is located at Yujiang Town, Yingtan City, Jiangxi Province, China (28°15′N,
116°55′E). The region has a typical subtropical monsoon climate with an annual mean temperature of
about 18 ℃ and an annual precipitation of about 1800 mm. Prior to the experiment, the field was
cultivated with early rice from April to July and late rice from July to November, and then kept in fallow
for the rest of year. The soil type at the experimental field is classified as Typical Haplaquepts (Soil
Survey Staff 1975). The initial properties (0–15 cm) of the soil are pH ($H_2O$) 4.74, organic carbon (SOC)
17.0 g $kg^{-1}$, and total N 1.66 g $kg^{-1}$. Daily air temperature (℃) and rainfall (mm) throughout the whole
observational period was provided by Red Soil Ecological Experiment Station, Chinese Academy of
Sciences (Appendix S1).
Four treatments, laid out in a randomized block design in triplicate, were conducted in the
experimental field after late-rice harvest from 2010 to 2014: (1) the plots were neither drainage nor
tillage in the whole winter fallow season as Treatment NTND, which is the traditional land management
in the local region; (2) the plots were drainage but non-tillage as Treatment NTD; (3) the plots were
tillage but non-drainage as Treatment TND; (4) and the plots were drainage and tillage simultaneously as
Treatment TD. Rice stubble in all treatments was around 25–35 cm long, about 3.0–4.0 t $ha^{-1}$ during the
4 winter fallow seasons, respectively. A small portion of rice stubble was collected before early-rice
transplanting and the total C and N contents were measured by the wet oxidation-redox titration method
and the micro-Kjeldahl method, respectively (Lu, 2000). Soil water content in winter fallow season was
determined gravimetrically after drying at 105 ℃ for 8 h.
Local rice (*Oryza sativa L.*) cultivars, Zhongzao 33 and Nongxiang 98, were planted for the following
early- and late-rice seasons, respectively. The seeds were sown in the seedling nursery and then
transplanted into the experimental plots at their 3- to 4-leaf stage. Each season, nitrogen (N) and
potassium (K) fertilizations in form of urea and potassium chloride (KCl) were split into three



applications, namely, basal fertilizers consisting of 90 kg N ha$^{-1}$ and 45 kg K ha$^{-1}$, tillering fertilizers
consisting of 54 kg N ha$^{-1}$ and 60 kg K ha$^{-1}$, and panicle initiation fertilizers consisting of 36 kg N ha$^{-1}$
and 45 kg K ha$^{-1}$. Phosphorus (P) fertilization in form of phosphorus pentoxide ($P_2O_5$) was applied to all
the treatments as basal fertilizer at a rate of 75 kg P ha$^{-1}$. Detailed descriptions about the water
management and fertilization are shown in Appendix S2.

**2.2 $CH_4$ and $N_2O$ fluxes sampling and measurements**
Both $CH_4$ and $N_2O$ fluxes were measured once every 2–6 d and 7–10 d during the rice and non-rice
seasons, respectively, using the static chamber technique (Zhang et al., 2011a). The flux chamber was 0.5
$\times 0.5 \times 1$ m, and plastic base (0.5 $\times 0.5$ m) for the chamber was installed before the experiment. Four gas
samples from each chamber were collected using 18-mL vacuum vials at 15-min intervals. Soil
temperature and soil redox potential (Eh) at 0.1 m depth were simultaneously measured during gas
collection. Rice grain yields were determined in each plot at early- and late-rice harvests.
The concentrations of $CH_4$ and $N_2O$ were analyzed with gas chromatographs equipped with a flame
ionization detector (Shimadzu GC-12A, Shimadzu Co., Japan) and with an electron capture detector
(Shimadzu GC-14B, Shimadzu Co., Japan), respectively. Both the emission fluxes were calculated from
the linear increase of gas concentration at each sampling time (0, 15, 30 and 45 min during the time of
chamber closure) and adjusted for area and volume of the chamber. Sample sets were rejected unless
they yielded a linear regression value of $r^2$ greater than 0.90. The amounts of $CH_4$ and $N_2O$ emissions
were calculated by successive linear interpolation of average $CH_4$ and $N_2O$ emissions on the sampling
days, assuming that $CH_4$ and $N_2O$ emissions followed a linear trend during the periods when no sample
was taken.

**2.3 GWPs and GHGI estimates**
The 100-year GWPs ($CH_4$ and $N_2O$) in different treatments were calculated by using IPCC factors
(100-year GWPs ($CH_4 + N_2O$) = 28 $\times CH_4$ + 265 $\times N_2O$) (Myhre, 2013). The greenhouse gas intensity
(GHGI) represented the GWPs per unit rice grain yield (Li et al., 2006): GHGI = GWPs/grain yield.

**2.4 Soil sampling and DNA extraction**
During the 2013–2014 winter fallow and early- and late-rice seasons, soil samples were collected in the



beginning, middle and end of each season from the experimental plots for analyzing the abundances of
methanogens and methanotrophs. Totally, there were 108 soil samples (3 seasons × 3 stages in each
season × 4 treatments × 3 replicates). Each sample was collected at 0–5 cm depth in triplicate and fully
mixed. Subsequently, all samples were stored at 4 ℃ for analyses of soil characteristics and subsamples
were maintained at −80 ℃ for DNA extraction.
For each soil sample, genomic DNA was extracted from 0.5 g soil using a FastDNA spin kit for soil
(MP Biomedicals LLC, Ohio, USA) according to the manufacturer's instructions. The extracted soil
DNA was dissolved in 50 μl of elution buffer, checked by electrophoresis on 1% agarose, and then
quantified using a spectrophotometer (NanoDrop Technologies, Wilmington, DE, USA) (Fan et al.,

160  2016).


**2.5 Real-time PCR quantification of *mcrA* and *pmoA* genes**
The abundance of methanogenic *mcrA* gene copies and of methanotrophic *pmoA* genes copies was
determined by quantitative PCR (qPCR) (Fan et al., 2016). Fragments of the *mcrA* and *pmoA* genes,
encoding the methyl coenzyme-M reductase and the α subunit of the particulate methane
monooxygenase, respectively, were amplified using primers according to Hales et al. (1996) and
Costello and Lidstrom (1999), respectively. Real-time quantitative PCR was performed on a CFX96
Optical Real-Time Detection System (Bio-Rad Laboratories, Inc. Hercules, USA), and for the detailed
descriptions please refer to our previous study (Fan et al., 2016).

**2.6 Statistical analyses**
Statistical analysis was performed using SPSS 18.0 software for Windows (SPSS Inc., USA).
Differences in seasonal $CH_4$ and $N_2O$ emissions, 100-year GWPs ($CH_4$ and $N_2O$), and grain yields
among treatments were analyzed with a repeated-measures one-way analysis of variance (ANOVA) and
least significant differences (LSD) test. The significance of the factors (land management and year) was
examined by using a two-way analysis of variance (ANOVA). Statistically significant differences and
correlations were set at $P < 0.05$.

**3 Results**
**3.1 $CH_4$ emission**



Obvious $CH_4$ fluxes were observed over the 4 winter fallow seasons, particularly during the 2011–2012
winter fallow season though a small net sink of $CH_4$ to the atmosphere was measured occasionally (Fig.
1). Total $CH_4$ emissions of the 4 treatments were highly lower ($P < 0.05$) in the 2010–2011 winter fallow
season (~0.1–1 kg $CH_4\,ha^{-1}$) than the following three winter fallow seasons (~1–11 kg $CH_4\,ha^{-1}$), and
they were ranged from 1.73 to 4.91 kg $CH_4\,ha^{-1}$ on average (Table 1). Seasonal $CH_4$ emissions varied
significantly with year and field managements (Table 2, $P < 0.01$). Tillage increased $CH_4$ emissions by
43–69% relative to non-tillage over the 4 winter fallow seasons. In comparison of non-drainage,
drainage reduced $CH_4$ emissions by 40–50%. Consequently, $CH_4$ emission was decreased by 14.8%
relative to Treatment NTND with the integrated effects of soil drainage and tillage (Table 1).
During the 4 early- and late-rice seasons, the $CH_4$ fluxes of all treatments dramatically ascended under
continuous flooding, and the highest $CH_4$ fluxes were observed on about 20–30 days after rice
transplanting in early-rice seasons and about 10–30 days after rice transplanting in late-rice seasons (Fig.
1). Subsequently, they sharply decreased after midseason aeration. An obvious flux peak was observed
again approximately 1–2 weeks after reflooding, particularly in the early-rice season. Apparently, the
$CH_4$ emission always showed a higher flux peak in Treatment NTND than in Treatment TD.
Seasonal $CH_4$ emissions in early-rice season varied significantly with land managements, but it was
not highly impacted by year or their interaction (Table 2). In contrast, total $CH_4$ emission did
significantly vary with land managements and year in late-rice season (Table 2). In comparison of
Treatment NTND, $CH_4$ emission was decreased by soil drainage and tillage, and on average, reduced by
22.2% and 17.8% in early- and late-rice seasons, respectively (Table 1). Soil drainage combined with
tillage further reduced $CH_4$ emission by 35.0% and 29.4% in early- and late-rice seasons, respectively.
Compared with early-rice season (68.3–105.1 kg $CH_4\,ha^{-1}$), total $CH_4$ emission in late-rice season was
8.0–17.9% greater.
Annually, total $CH_4$ emission was ranged from 151 to 222 kg $CH_4\,ha^{-1}$, averaged 46.1% and 52.1% of
which came from the early- and late-rice seasons, respectively (Tables 1 and 3). Soil drainage and tillage
played important roles in decreasing $CH_4$ emission. Relative to Treatment NTND, averaged $CH_4$
emission was decreased by 24.3% and 14.9% by drainage and tillage, separately, and it was highly
reduced by 32.0% when drainage was combined with tillage simultaneously (Table 3).

**3.2 $N_2O$ emission**





Substantial $N_2O$ emission was measured in the non-rice growth season though the fields were fallowed
with no N-fertilization (Fig. 2 and Table 1). Total $N_2O$ emissions over the 4 winter fallow seasons varied
significantly with land management and year while it did not significantly depended on their interaction
(Table 2). Seasonal $N_2O$ emissions were relatively lower in the 2010–2012 winter fallow seasons than
the following two winter fallow seasons. Compared with Treatment NTND, soil drainage and tillage
generally increased $N_2O$ emissions, separately, and $N_2O$ emissions were significantly stimulated when
combined drainage with tillage simultaneously. Over the 4 winter fallow seasons, seasonal $N_2O$
emissions averaged 36.4–68.2 g $N_2O$–N $ha^{-1}$, being 87.3%, 64.5% and 57.5% higher in Treatment TD
than in Treatments NTND, TND, and NTD, respectively (Table 1).
After rice transplanting, pronounced $N_2O$ fluxes were observed with N-fertilization and midseason
aeration, particularly in the period of dry/wet alternation (Fig. 2). Two-way ANOVA analyses indicated
that seasonal $N_2O$ emissions during the early- and late-rice seasons were not highly influenced by land
management, and the interactions of land management and year, except that $N_2O$ emissions depended
significantly on year (Table 2). Compared with Treatments NTND and NTD, tillage increased $N_2O$
emission in 2011 early- and late-rice seasons whereas generally reduced $N_2O$ emission during the
following rice seasons (Table 1).
Over the 4 early-rice seasons, drainage increased seasonal $N_2O$ emissions by 38.9–43.5% while tillage
decreased by 10–12.9%, although no significant difference was observed (Table 1). In contrast, the
effects of drainage and tillage seemed to be more important over the 4 late-rice seasons. For instance,
drainage increased seasonal $N_2O$ emissions by 41.0–47.8% while tillage decreased by 10.3–14.4%.
Annually, total $N_2O$ emission was ranged from 113 to 167 g $N_2O$-N $ha^{-1}$, averaged 34.4% of which
derived from the winter fallow season (Tables 1 and 3). There was no significant difference in total $N_2O$
emission among the 4 treatments (Table 3).

**235 3.3 Global warming potential (GWP)**

Throughout the 4 winter fallow seasons, soil drainage and tillage had important effects on GWPs over
the 100-year time, although it was, on average, very small, being from 0.07 to 0.16 t $CO_2$-eq $ha^{-1}$ $yr^{-1}$
(Table 1). Compared with Treatment NTND, drainage significantly decreased GWPs while tillage highly
increased it. Consequently, soil drainage combined with tillage played a slightly role in GWPs relative to
Treatment NTND.





In contrast, both soil drainage and tillage decreased GWPs in comparison of Treatment NTND over
the 4 early-rice seasons, with 16.0–36.2% and 4.2–36.2% lower in Treatment NTD and Treatment TND,
respectively (Table 1). The GWPs was hence far more decreased by drainage combined with tillage,
being 26.6–42.4% lower in Treatment TD than in Treatment NTND. Totally, drainage significantly
reduced GWPs by 27.4% for Treatment NTD, in particular on Treatment TD by 34.8% with the
integrated effect of drainage and tillage relative to Treatment NTND. Meanwhile, tillage tended to
decrease GWPs relative to Treatment NTND but this effect was not statistically significant.
Similar effects of soil drainage and tillage on GWPs were observed over the 4 late-rice seasons (Table
1). Compared with Treatment NTND, GWPs was 7.5–35.4% and 11.7–20.4% lower in Treatments NTD
and TND, respectively. Soil drainage combined with tillage significantly decreased GWPs by
23.7–36.8% for Treatment TD in comparison of Treatment NTND. On average, drainage and tillage
reduced GWPs by 20.6% and 15%, separately, and GWPs was significantly reduced (29.1%) by
combining drainage with tillage simultaneously.
Annually, the GWPs averaged 4.29–6.25 t $CO_2$-eq ha$^{-1}$, with 46% and 52% of which derived from the
early-rice and late-rice seasons, respectively (Tables 1 and 3). Compared with Treatment NTND, GWPs
was significantly reduced by 0.92–1.49 t $CO_2$-eq ha$^{-1}$ in Treatments TND and NTD, respectively, and it
was decreased much more by 1.96 t $CO_2$-eq ha$^{-1}$ in Treatment TD (Table 3).

**3.4 Rice grain yields**
Grain yields of Treatments TND and TD are generally higher than those of Treatments NTND and NTD
over the 4 annual cycles (Table 1) though the yields slightly varied with land management and year as
well as their interaction (Table 2). On average, the yields in Treatments TND and TD were over 6.5 t ha$^{-1}$,
4.8%–7.3% and 3.1%–4.4% higher than those of Treatments NTND and NTD during the early- and
late-rice seasons, respectively. Annually, no significance in the total yields was observed among the
treatments over the 4 years (Table 3). Throughout the 4 late-rice seasons, positive correlation was
observed between grain yields of 4 treatments and the corresponding $CH_4$ emissions (r= 0.733, $P <$

267    0.01).


**3.5 Greenhouse gas intensity (GHGI)**
Annual GHGI ranged from 0.32 to 0.49 t $CO_2$-eq t$^{-1}$ yield, and it changed significantly among the





treatments owing to GWPs highly controlled while annual rice yields slightly influenced by soil drainage
and tillage (Table 3). Compared with Treatment NTND, drainage and tillage reduced GWPs by 23.8%
and 14.7%, thus causing GHGI significantly decreased by 22.4% and 18.4%, separately. Expectedly, soil
drainage combined with tillage reduced GHGI much more, with a reduction of 34.7% relative to
Treatment NTND.

**3.6 Precipitation, temperature, soil Eh and soil water content in winter fallow season**
Over the 4 winter fallow seasons, total precipitation changed remarkably, which was ranged from ~400
mm to ~750 mm during 2010–2012. Subsequently, it was relatively stable around 600 mm in 2012–2014
(Table 4). In contrast, mean daily air temperature varied slightly, with values of ca. 9.0 ℃ to 10.0 ℃.
Soil Eh, on average, fluctuated obviously from the highest (~150 mV) in 2010–2011 to the lowest (~90
mV) in 2013–2014. Soil water content in 2010 winter fallow season was generally higher in Treatment
NTND than in Treatments NTD and TND, and it was lowest in Treatment TD (Fig. 3a), with a mean
value of 55%, 50%, 44% and 38%, respectively. It is easy to see that the higher the precipitation and
temperature, the lower the soil Eh, and thus the more the $CH_4$ emission in winter fallow season (Table 4).
Statistical analyses show that a significant exponential relationship was observed between mean $CH_4$
emission and total precipitation (Fig. 3b, $P < 0.01$), and mean $CH_4$ emission positively and negatively
correlated with mean temperature (Fig. 3c, $P < 0.05$) and soil Eh (Fig. 3d, $P < 0.01$), respectively.

**3.7 Abundance of methanogens and methanotrophs populations**
The abundance of methanogens in paddy soil decreased significantly from winter fallow season to the
following early-rice season, but it increased again during the late-rice season (Fig. 4a). Compared with
non-drainage (Treatments NTND and TND), drainage (Treatments NTD and TD) generally decreased
the abundance of methanogens throughout the winter fallow (Fig. 4a, $P < 0.001$) and following early-
and late-rice seasons (Fig. 4a, $P < 0.05$). Relative to non-tillage (Treatments NTND and NTD), tillage
(Treatments TND and TD) also significantly decreased the abundance of methanogens throughout the
winter fallow and following early- and late-rice seasons (Fig. 4a, $P < 0.001$).
The abundance of methanotrophs was highest in winter fallow season, and then it decreased gradually
(Fig. 4b). Drainage (Treatments NTD and TD) relative to non-drainage (Treatments NTND and TND)
significantly decreased the abundance of methanotrophs over the winter fallow and early-rice seasons



(Fig. 4b, $P < 0.05$) though no significance during the late-rice season. In addition, tillage (Treatments
TND and TD) significantly decreased the abundance of methanogens during the previous winter (Fig. 4b,
$P < 0.001$) and following early-rice seasons (Fig. 4b, $P < 0.01$) in comparison of non-tillage (Treatments
NTND and NTD), except in the late-rice season.

**4 Discussion**
**4.1 $CH_4$ emission from double-rice fields**
It is reported that *in situ* measurement of $CH_4$ emission in China was firstly carried out from 1987 to
1989 in a double-rice field in Hangzhou City (Shangguan et al., 1993b). Subsequently, more and more
$CH_4$ emissions from double-rice fields were observed (Cai et al., 2001; Shang et al., 2011). However,
few investigations were referred to related measurements in the non-rice growth season. Fortunately,
Shang et al. (2011) found the double-rice fields in Hunan province China usually acting as a small net
sink of $CH_4$ emission (as low as $-6$ kg $CH_4$ ha$^{-1}$) in winter fallow season. Although an occasionally
negative $CH_4$ flux was also observed over the 4 winter fallow seasons (Fig. 1), the double-rice field in
this study was an entire source of $CH_4$ emission, in particular during the 2011–2012 winter fallow season
(Table 1). On average, around 2% of annual $CH_4$ emission emitted from the winter fallow season.
Because of the residues (mainly including roots and stubble) of early rice as well as high temperature
resulting in substantial $CH_4$ production in paddy fields (Shangguan et al., 1993a; Yan et al., 2005), $CH_4$
emission of late-rice season was generally higher than that of early-rice season. More importantly, a very
high $CH_4$ flux peak was usually observed in a couple of days after late-rice transplanting (Cai et al., 2001;
Shang et al., 2011). In the present study, $CH_4$ emission in late-rice seasons was 80.1–113.5 kg $CH_4$ ha$^{-1}$,
being 8.0–17.9% larger than that of early-rice seasons (Table 1) though total $CH_4$ emission in the last two
early-rice seasons was found to be slight greater than those in late-rice seasons (Fig. 1). Mean annual
$CH_4$ emission varied between 151 and 222 kg $CH_4$ ha$^{-1}$ over the 4 years (Table 3), which was much
lower than previous results (Cai et al., 2001; Shang et al., 2011). Great differences in these $CH_4$
measurements were probably attributed to different water and rice straw managements.
Significant differences in $CH_4$ emission from the fields in winter fallow and late-rice seasons were
observed (Table 2), indicating large changes in the interannual $CH_4$ emission. It is believed that the
climatic variation may be the major factor leading to interannual variation of $CH_4$ emission at the
macroscopic scale (Cai et al., 2009). In this study we found that total winter rainfall had an important





effect on $CH_4$ emission, and the higher the rainfall, the greater the $CH_4$ emission throughout the 4 winter
fallow seasons (Table 4). And an exponential relationship was observed between mean $CH_4$ emission and
total rainfall in winter fallow season (Fig. 3b). The importance of rainfall in controlling $CH_4$ emission in
winter fallow season, to some extent, also could be demonstrated by the negative relationships between
mean soil Eh and $CH_4$ emission (Fig. 3d). According to different rice fields from 4 main rice growing
regions in China, similar correlation was found between rainfall in winter fallow season and $CH_4$
emission in the rice growth season (Kang et al., 2002).

338         Nevertheless, we did not found any correlations between rainfall in winter fallow season and $CH_4$ flux

in early-or late-rice season in this study, suggesting that rainfall in winter fallow season just significantly
regulated $CH_4$ flux on-season, but didn't off-season. In contrast, a significant linear relationship was
found ($P < 0.01$) between $CH_4$ emissions and corresponding yields over the 4 late-rice seasons,
demonstrating that crop growth benefited rice yield and biomass and thus stimulated $CH_4$ emission. It is
reported that seasonal $CH_4$ emission depended greatly on rice biomass based on a long-term fertilizer
experiment (Shang et al., 2011). Furthermore, changes in temperature over the 4 winter fallow seasons
(Table 4) were supposed to play a key role in $CH_4$ emission, and the positive correlation had
demonstrated this well (Fig. 3c). Many field measurements have shown the importance of temperature to
$CH_4$ emission (Cai et al., 2003; Parashar et al., 1993; Zhang et al., 2011a).

**4.2 Effect of soil drainage in winter fallow season on $CH_4$ emission**
Considerable measurements of $CH_4$ emission as affected by soil drainage in winter fallow season have
been reported from single-rice fields, and most of which were from the permanently flooded fields.
Obviously, drainage significantly decreases $CH_4$ emission (Table 5). Draining the flooded fields inhibits
$CH_4$ production and $CH_4$ emission in winter fallow season directly, and more importantly, it plays an
important role in reducing $CH_4$ production and its emission in the subsequent rice-growing season
(Zhang et al., 2011a). Compared with non-drainage, drainage in this study significantly decreased $CH_4$
emission both in previous winter fallow seasons and following early- and late-rice seasons (Table 1), and
over the 4 years, mean annual $CH_4$ emission was reduced by 38–54 kg $CH_4$ $ha^{-1}$ (Table 3). Such changes
were very likely due to the decrease of methanogens in paddy soils throughout the winter, early- and
late-rice seasons by soil drainage (Fig. 4a) because drainage increases soil aeration and hence effectively
reduces the survival rate and activity of methane-producing bacteria. According to microcosm





experiments, Ma and Lu (2011) found that the total abundance of methanogenic archaeal populations
decreased by 40% after multiple drainages, and quantitative PCR analysis further revealed that both mcrA
gene copies and mcrA transcripts significantly decreased after dry/wet alternation (Ma et al., 2012).

**4.3 Effect of soil tillage in winter fallow season on $CH_4$ emission**
Although $CH_4$ emission in winter fallow season was increased by soil tillage, it was highly decreased
during the following early- and late-rice seasons (Table 1), and over the 4 years, on average, it was
reduced by 17–33 kg $CH_4$ $ha^{-1}$ $yr^{-1}$ (Table 3). Compared to non-tillage, tillage may promote the
decomposition of rice residues, and then stimulates $CH_4$ production and emission in winter fallow season.
By contrast, as the readily decomposable part of the residues has largely been decomposed after a whole
winter fallow season, the remaining hardly-decomposable part of organic matter doesn't have much
effect on promoting $CH_4$ emission next year (Watanabe and Kimura, 1998). The content of total C in rice
residues generally lower in Treatments TND and TD than in Treatments NTND and NTD (Table 6) has
well demonstrated that tillage decreased the carbon substrates for methanogenesis. It therefore, relative
to non-tillage, significantly reduced $CH_4$ emission (Table 3). In a rice-wheat rotation system, our 2-year
field measurements also showed that the carbon content of rice straw incorporated into the soil in winter
fallow season was decreased sharply in comparison of that applied to the field just prior to rice
transplanting (Zhang et al., 2015). In addition, tillage highly reduced the abundance of methanogens
throughout the winter fallow and early- and late-rice seasons (Fig. 4a) should be a probable reason for
the decrease of $CH_4$ emission.

**4.4 $N_2O$ emission from double-rice paddy fields**
Direct $N_2O$ emission from rice-based ecosystems mainly happens in the periods of midseason aeration
and subsequent dry/wet alternation in rice-growing season, and in winter crop or fallow season (Zheng et
al., 2004; Cai et al., 1997; Ma et al., 2013; Yan et al., 2003). It is estimated that most of croplands $N_2O$
emission comes from uplands and just 20–25% of which is from rice fields in China (Zhang et al., 2014).
In China, field measurements of $N_2O$ emission began in 1992 from a single-rice field in Liaoning
province (Chen et al., 1995), and considerable observations from double-rice fields had been performed
(Xu et al., 1997; Shang et al., 2011; Zhang et al., 2013a). The total $N_2O$ emission of early- and late-rice
seasons in this study, on average, varied between 70.6 and 114.7 g $N_2O$-N $ha^{-1}$ $yr^{-1}$ over the 4 years




(Table 1), being significantly lower than those reported by Shang et al. (2011) and Zhang et al. (2013a)
but similar to our previous measurements Ma et al. (2013). Furthermore, over 1/3 of annual $N_2O$
emission came from the winter fallow season (Table 1), indicating that $N_2O$ emission from paddy fields
in winter fallow season was very important. Early field observations even showed that as high as
60–90% of $N_2O$ emission occurred in winter fallow season (Shang et al., 2011). On a national scale, it is
found that 41 Gg $N_2O$-N $yr^{-1}$ emitted in the non-rice growth period, contributing 45% of the total $N_2O$
emission from rice-based ecosystems (Zheng et al., 2004). Although $N_2O$ emission from rice fields
significantly affected by year (Table 2), reasons for the interannual variation were still not well known.
In order to specify rules for interannual change in $N_2O$ emission, it is essential to maintain
all-the-year-round long-term stationary field observations of $N_2O$ emission from the double-rice fields.

**4.5 Effect of soil drainage in winter fallow season on $N_2O$ emission**
The production of soil $N_2O$ is mainly by the microbial processes of nitrification and denitrification while
soil water content determines the general direction of the transformation of soil nitrogen. Soil drainage
can cut down the soil water content and accelerate soil dry/wet alternation, thus promoting $N_2O$ emission
from paddy fields (Davidson, 1992; Cai et al., 1997). It is because that soil dry/wet alternation stimulates
the transformation of C and N in the soil, in particular on the microbial biomass C and N turnover
(Potthoff et al., 2001). Expectedly, drainage usually decreased the soil water content in this study (Fig. 3a)
and then increased $N_2O$ emission, on average, by 42% relative to non-drainage in winter fallow season
(Table 1). Noted that drainage in previous winter fallow season also had an important effect on $N_2O$
emission from paddy fields during the following rice seasons, namely, it increased $N_2O$ emission both in
early- and late-rice seasons (Table 1). It was possibly attributed to that drainage in winter fallow season
would create soil moisture more beneficial to $N_2O$ production in the subsequent rice-growing seasons.
Early report had well demonstrated that the production and emission of soil $N_2O$ was not only related to
the soil moisture regime at the time, but also strongly affected by the previous soil moisture regime
(Groffman and Tiedje, 1988). And regardless of how the water conditions were at that time, the previous
soil moisture conditions affected the concentration of reductase or synthetic ability of the enzymes, thus
affecting denitrification (Dendooven and Anderson, 1995; Dendooven et al., 1996). Totally, annual $N_2O$
emission was increased by 37–48% compared drainage with non-drainage though there was no
significant difference among the 4 treatments (Table 3).






**4.6 Effect of soil tillage in winter fallow season on N$_2$O emission**

Compared to non-tillage, tillage usually increased N$_2$O emission in winter fallow season, on average, by

39% over the 4 years (Table 1), which might be ascribed to two reasons. First, tillage increases soil

aeration, which possibly promotes the process of nitrification. A soil column experiment has well

demonstrated that moderate O$_2$ concentration is conducive to N$_2$O production (Khdyer and Cho, 1983).

Second, tillage accelerates rainwater from the plow layer percolating into the subsoil layer, stimulating

the processes of soil dry/wet alternation and then promoting the transformation of N and production of

N$_2$O in the soil (Cai et al., 1997; Potthoff et al., 2001). Tillage usually decreased soil water content (Fig.

3a) could validate this to some extent. In contrast, it had negative effects on N$_2$O emission during the

following early- and late-rice seasons, and mean N$_2$O emission over the 4 years was reduced by 12% and

13%, respectively (Table 1). Compared to non-tillage, tillage decreased the content of total N in rice

residues, which probably reduced the substrates for nitrification and denitrification. More importantly,

the ratio of C/N in rice residues was increased by tillage (Table 6). Because the decomposition of rice

residues with high C/N ratio probably resulted in more N immobilization in the soil and less N available

to nitrification and denitrification for N$_2$O production (Huang et al., 2004; Zou et al., 2005). As a whole,

soil tillage played a slight role in annual N$_2$O emission over the 4 years (Table 3).

438

**4.7 Effect of soil drainage and tillage on GWPs and GHGI**

Although drainage increased N$_2$O emission throughout the winter fallow, and early- and late-rice seasons,

it significantly decreased CH$_4$ emission from paddy fields (Table 1). As a consequence, it highly reduced

GWPs, with a decrease of 1.49 t CO$_2$-eq ha$^{-1}$ annually (Table 3). Considerable studies have showed that

drainage results in a trade-off between CH$_4$ and N$_2$O emissions from rice fields (Table 5), and it is widely

considered to be an effective mitigation option. Annually, the mitigation potential of GWPs from paddy

fields by drainage in winter fallow season is over 50%. However, these measurements are mostly related

to the single-rice fields with continuous flooding (Table 5), and few information are available about the

effect on GWPs from double rice-cropping systems. In this study, we found that as high as 21–30% of

the GWPs reduced by drainage in winter fallow season throughout the previous winter fallow and

following early- and late-rice seasons, and with 24% of mitigation potential annually (Table 3).

In contrast, tillage obviously increased both CH$_4$ and N$_2$O emissions, thus highly increased GWPs in



winter fallow season (Table 1). Indeed, in a single-rice field, Liang et al. (2007) found that it increased
the GWPs of $CH_4$, $N_2O$ and $CO_2$ emissions in winter fallow season (Table 5). Fortunately, it significantly
decreased $CH_4$ and $N_2O$ emissions both in early-and late-rice seasons, and as a result, with a reduction of
GWPs by 17% and 15%, respectively (Table 1). Annually, the GWPs were reduced by 0.92 t $CO_2$-eq
$ha^{-1}$, with 15% of mitigation potential (Table 3). As expected, the integrated effects of soil drainage and
tillage decreased GWPs much more, with a further reduction by 1.04 t $CO_2$-eq $ha^{-1}$ $yr^{-1}$. Moreover, the
annual mitigation potential (as high as 32%) of soil drainage combined with tillage in this study was in
the ranges of previous results reported by Zhang et al. (2012) and Zhang et al. (2015) in single-rice fields
(Table 5). It is obvious that the soil drainage together with tillage simultaneously in winter fallow season
might be an effective option for mitigating the GWPs of $CH_4$ and $N_2O$ emissions from the double
rice-cropping systems.
More importantly, no significant difference in rice grain yields was observed among the 4 treatments
over the 4 years (Tables 1 and 3). It suggests that we would not risk rice yield loss when we try to
decrease the GWPs of $CH_4$ and $N_2O$ emissions by means of soil drainage or tillage in winter fallow
season. So, soil drainage and tillage significantly decreased GHGI by 22.4% and 18.4%, separately, and
the GHGI was decreased much more by combining drainage with tillage, with a reduction of 0.17 t
$CO_2$-eq $t^{-1}$ yield $yr^{-1}$ (Table 3). Based on a long-term fertilizer experiment, balanced fertilizer
management, in particular on P fertilizer supplement, was suggested to be an available strategy in double
rice-cropping systems (Shang et al., 2011). In this study, the effective mitigation option in double-rice
fields we proposed is that soil drainage combined with tillage in winter fallow season.
In Conclusion, the study demonstrated that in winter fallow season large differences in $CH_4$ emissions
were probably due to the changes in total precipitation and temperature. Soil drainage and tillage in
winter fallow season separately, in particular on combining both of them, significantly decreased $CH_4$
emission and then GWPs of $CH_4$ and $N_2O$ emissions from double-rice field. One possible explanation for
this phenomenon is that drainage and tillage decreased the abundance of methanogens in paddy soil.
Moreover, low total C content in rice residues due to tillage was a potential reason for the decrease of
$CH_4$ emission in the following early- and late-rice seasons. Finally, tillage reduced total N content but
increased C/N ratio in rice residues would be important to the decrease of $N_2O$ emission. For both
achieving high rice grain yield and low GWPs in double-rice fields, land management strategies in this
study we proposed, including the fields were drained immediately after late-rice harvest, and meanwhile,





the fields were tilled with rice residues incorporated into the soil. The results would benefit the
development of optimal management strategies in the double-rice systems and the interpretation of the
corresponding mechanisms.

**Acknowledgements**
This work was financially supported by the "Strategic Priority Research Program" of
the Chinese Academy of Sciences (XDB15020103), the National Key Technology Research and
Development Program (2013BAD11B02), the National Natural Sciences Foundation of China
(41571232, 41271259), Foundation of the State Key Laboratory of Soil and Sustainable Agriculture
(Y412010003), and the Knowledge Innovation Program of Institute of Soil Science, Chinese Academy
of Sciences (ISSASIP1654). We sincerely thank Red Soil Ecological Experiment Station, Chinese
Academy of Sciences for providing climate information. Deep appreciation also goes to the anonymous
reviewers for their helpful comments.

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





**Figure captions:**

**Figure 1** Seasonal variation of $CH_4$ emission from 2010 to 2014.


**Figure 2** Seasonal variation of $N_2O$ emission from 2010 to 2014.


**Figure 3** Soil water content in 2010 winter fallow season (a) and the relationships
between mean $CH_4$ emission and total winter precipitation (b), and mean daily air
temperature (c) and soil Eh (d) over the 4 winter fallow seasons (Data from Table 4).


**Figure 4** The abundance of methanogens and methanotrophs populations in paddy soil
from 2013 to 2014, WS, ES, and LS means winter fallow season, early-rice season, and
late-rice season, respectively.








**Table 1** Seasonal CH₄ and N₂O emissions, global warming potentials (GWPs), and rice grain yields over the 4 years from 2010 to 2014.

| Year | Treatment | Winter fallow season | | | Early-rice season | | | | Late-rice season | | | |
|---|---|---|---|---|---|---|---|---|---|---|---|---|
| | | $CH_4$ emission (kg $CH_4$ ha⁻¹) | $N_2O$ emission (g $N_2O$-N ha⁻¹) | GWPs (t $CO_2$-eq ha⁻¹) | $CH_4$ emission (kg $CH_4$ ha⁻¹) | $N_2O$ emission (g $N_2O$-N ha⁻¹) | GWPs (t $CO_2$-eq ha⁻¹) | Yield (t ha⁻¹) | $CH_4$ emission (kg $CH_4$ ha⁻¹) | $N_2O$ emission (g $N_2O$-N ha⁻¹) | GWPs (t $CO_2$-eq ha⁻¹) | Yield (t ha⁻¹) |
| 2010–2011 | TD | 0.46 ±0.02 | 46.4 ±1.5 | 0.03 ±0.01 | 61.3 ±12.5 | 49.0 ±7.2 | 1.74 ±0.39 | 6.44 ±0.82 | 133.9 ±18.6 | 98.5 ±4.3 | 3.79 ±0.17 | 7.13 ±0.07 |
| | TND | 1.05 ±0.13 | 30.4 ±3.1 | 0.04 ±0.02 | 80.6 ±2.4 | 46.6 ±7.1 | 2.28 ±0.06 | 6.29 ±0.20 | 158.5 ±28.3 | 67.4 ±2.1 | 4.46 ±0.40 | 7.33 ±0.09 |
| | NTD | 0.11 ±0.19 | 42.7 ±5.3 | 0.02 ±0.02 | 70.6 ±6.1 | 45.3 ±11.1 | 2.00 ±0.16 | 6.08 ±0.60 | 147.0 ±15.6 | 62.8 ±5.1 | 4.14 ±0.02 | 6.72 ±0.22 |
| | NTND | 0.38 ±0.07 | 32.2 ±5.1 | 0.02 ±0.01 | 84.9 ±14.3 | 38.9 ±12.3 | 2.38 ±0.29 | 5.82 ±0.34 | 179.6 ±26.2 | 44.5 ±11.0 | 5.05 ±0.15 | 6.83 ±0.84 |
| 2011–2012 | TD | 5.06 ±1.18 | 42.0 ±1.8 | 0.16 ±0.04 | 64.0 ±12.5 | 17.7 ±7.9 | 1.80 ±0.35 | 6.67 ±0.08 | 79.6 ±8.8 | 45.2 ±7.8 | 2.25 ±0.24 | 6.63 ±0.09 |
| | TND | 11.1 ±2.51 | 35.1 ±2.7 | 0.33 ±0.07 | 90.6 ±8.2 | 16.2 ±7.2 | 2.54 ±0.23 | 7.03 ±0.50 | 103.1 ±6.0 | 35.4 ±8.0 | 2.90 ±0.16 | 6.70 ±0.21 |
| | NTD | 4.54 ±0.32 | 27.3 ±11.3 | 0.14 ±0.04 | 68.1 ±11.8 | 28.2 ±6.1 | 1.92 ±0.22 | 6.36 ±0.36 | 81.0 ±4.3 | 63.0 ±9.6 | 2.30 ±0.80 | 6.57 ±0.35 |
| | NTND | 7.09 ±1.08 | 14.1 ±4.4 | 0.20 ±0.05 | 107.1 ±9.9 | 23.4 ±4.8 | 3.01 ±0.27 | 6.67 ±0.47 | 126.4 ±12.2 | 47.2 ±11.0 | 3.56 ±0.66 | 6.53 ±0.14 |
| 2012–2013 | TD | 1.40 ±0.21 | 88.2 ±14.7 | 0.08 ±0.02 | 79.7 ±15.2 | 27.5 ±4.1 | 2.24 ±0.49 | 6.33 ±0.50 | 44.3 ±2.1 | 32.3 ±3.7 | 1.25 ±0.07 | 6.46 ±0.41 |
| | TND | 3.75 ±0.21 | 59.7 ±18.0 | 0.13 ±0.02 | 101.1 ±14.8 | 17.7 ±15.0 | 2.84 ±0.42 | 6.48 ±0.78 | 52.7 ±11.1 | 15.3 ±3.5 | 1.48 ±0.31 | 6.30 ±0.23 |
| | NTD | 0.73 ±0.22 | 52.0 ±9.1 | 0.04 ±0.01 | 80.6 ±9.6 | 36.4 ±13.1 | 2.27 ±0.27 | 6.05 ±0.47 | 60.8 ±11.8 | 38.1 ±2.4 | 1.72 ±0.34 | 6.27 ±0.50 |
| | NTND | 2.11 ±0.23 | 56.5 ±13.0 | 0.08 ±0.00 | 108.7 ±5.8 | 24.1 ±14.9 | 3.05 ±0.15 | 6.38 ±0.73 | 65.9 ±12.9 | 32.3 ±6.7 | 1.86 ±0.36 | 6.08 ±0.24 |
| 2013–2014 | TD | 2.94 ±0.78 | 96.1 ±22.9 | 0.12 ±0.04 | 68.1 ±7.0 | 76.0 ±15.1 | 1.94 ±0.29 | 7.07 ±0.34 | 62.6 ±4.7 | 49.5 ±2.8 | 1.77 ±0.14 | 6.64 ±0.31 |
| | TND | 3.73 ±0.85 | 44.7 ±26.0 | 0.12 ±0.08 | 76.2 ±5.0 | 42.1 ±8.0 | 2.15 ±0.11 | 6.43 ±0.60 | 72.1 ±9.2 | 42.1 ±12.9 | 2.04 ±0.25 | 6.38 ±0.47 |
| | NTD | 1.52 ±0.48 | 52.0 ±28.4 | 0.06 ±0.02 | 88.4 ±6.3 | 85.4 ±10.9 | 2.51 ±0.21 | 6.19 ±0.23 | 70.6 ±13.6 | 99.7 ±7.5 | 2.02 ±0.39 | 6.46 ±0.61 |
| | NTND | 2.01 ±0.09 | 42.9 ±10.6 | 0.07 ±0.04 | 119.7 ±10.8 | 49.4 ±13.6 | 3.37 ±0.33 | 6.16 ±0.36 | 82.2 ±3.1 | 54.4 ±9.5 | 2.32 ±0.08 | 6.16 ±0.12 |
| Mean* | TD | 2.47 ±0.10 bc | 68.2 ±16.4 a | 0.10 ±0.02 b | 68.3 ±11.4 b | 42.5 ±11.2 a | 1.93 ±0.32 b | 6.62 ±0.25 a | 80.1 ±2.7 c | 56.4 ±17.4 ab | 2.27 ±0.08 c | 6.71 ±0.14 a |
| | TND | 4.91 ±0.43 a | 42.5 ±12.3 ab | 0.16 ±0.02 a | 87.2 ±13 ab | 30.6 ±15.0 a | 2.45 ±0.37 ab | 6.56 ±0.49 a | 96.6 ±8.3 b | 40.0 ±4.3 b | 2.72 ±0.23 b | 6.68 ±0.24 a |
| | NTD | 1.73 ±0.37 c | 43.5 ±18.4 ab | 0.07 ±0.00 c | 76.2 ±6.9 b | 48.8 ±18.1 a | 2.15 ±0.19 b | 6.17 ±0.27 a | 89.9 ±1.2 bc | 65.9 ±6.6 a | 2.54 ±0.03 bc | 6.51 ±0.39 a |
| | NTND | 2.90 ±0.21 b | 36.4 ±13.5 b | 0.10 ±0.02 b | 105.1 ±15.5 a | 34.0 ±6.9 a | 2.96 ±0.44 a | 6.26 ±0.33 a | 113.5 ±8.0 a | 44.6 ±8.0 b | 3.20 ±0.22 a | 6.40 ±0.20 a |

Mean* ±SD, different letters within the same column indicate statistical differences in variables mean among treatments over the 4 years by LSD's multiple range test ($P < 0.05$).








**Table 2** A two-way ANOVA for the effects of land management (L) and year (Y) on
$CH_4$ and $N_2O$ emissions and grain yields in the rice field.

| Season | Factors | df | $CH_4$ (kg $CH_4$ ha$^{-1}$) | | | $N_2O$ (g $N_2O$-N ha$^{-1}$) | | | Yield (t ha$^{-1}$) | | |
|---|---|---|---|---|---|---|---|---|---|---|---|
| | | | ss | F | P | ss | F | P | ss | F | P |
| Early-rice | L | 3 | 3052.7 | 5.196 | 0.005 | 820.1 | 1.007 | 0.403 | 0.603 | 2.361 | 0.090 |
| | Y | 3 | 692.3 | 1.178 | 0.333 | 4357.4 | 5.349 | 0.004 | 0.598 | 3.340 | 0.092 |
| | L × Y | 9 | 254.2 | 0.433 | 0.907 | 267.0 | 0.328 | 0.959 | 0.161 | 0.631 | 0.762 |
| | Model | 15 | 901.5 | 1.535 | 0.151 | 1195.7 | 1.468 | 0.176 | 0.337 | 1.319 | 0.248 |
| | Error | 32 | 587.5 | | | 814.7 | | | 0.256 | | |
| Late-rice | L | 3 | 2379.4 | 4.700 | 0.008 | 1635.2 | 1.528 | 0.226 | 0.259 | 1.522 | 0.228 |
| | Y | 3 | 22545.7 | 44.534 | 0.000 | 3515.8 | 3.286 | 0.033 | 1.193 | 7.015 | 0.001 |
| | L × Y | 9 | 223.0 | 0.440 | 0.903 | 826.9 | 0.806 | 0.614 | 0.057 | 0.338 | 0.955 |
| | Model | 15 | 5118.8 | 10.111 | 0.000 | 1547.9 | 1.447 | 0.185 | 0.325 | 1.910 | 0.061 |
| | Error | 32 | 506.3 | | | 1070.0 | | | 0.170 | | |
| Winter | L | 3 | 21.582 | 5.215 | 0.005 | 2367.6 | 4.537 | 0.009 | | | |
| | Y | 3 | 86.036 | 20.788 | 0.000 | 3265.9 | 6.259 | 0.002 | | | |
| | L × Y | 9 | 4.020 | 0.971 | 0.481 | 314.4 | 0.603 | 0.785 | | | |
| | Model | 15 | 23.935 | 5.783 | 0.000 | 1315.4 | 2.521 | 0.014 | | | |
| | Error | 32 | 4.139 | | | 521.8 | | | | | |























**Table 3** Mean annual $CH_4$ and $N_2O$ emissions, global warming potentials (GWPs) of
$CH_4$ and $N_2O$ emissions, rice grain yields, and greenhouse gas intensity (GHGI) over
the 4 years from 2010 to 2014.

| Treatment | $CH_4$ emission (kg $CH_4$ ha$^{-1}$ yr$^{-1}$) | $N_2O$ emission (g $N_2O$-N ha$^{-1}$ yr$^{-1}$) | GWPs (t $CO_2$-eq ha$^{-1}$ yr$^{-1}$) | Rice yields (t ha$^{-1}$ yr$^{-1}$) | GHGI (t $CO_2$-eq t$^{-1}$ yield yr$^{-1}$) |
|---|---|---|---|---|---|
| TD | 151 ±10 d | 167 ±28 a | 4.29 ±0.27 d | 13.3 ±0.3 a | 0.32 ±0.02 c |
| TND | 189 ±15 b | 113 ±13 a | 5.33 ±0.41 b | 13.2 ±0.6 a | 0.40 ±0.05 b |
| NTD | 168 ±6 cd | 158 ±27 a | 4.76 ±0.17 cd | 12.7 ±0.6 a | 0.38 ±0.02 b |
| NTND | 222 ±9 a | 115 ±38 a | 6.25 ±0.26 a | 12.7 ±0.1 a | 0.49 ±0.02 a |

Note: different letters within the same column indicate statistical differences among
treatments at $P < 0.05$ level by LSD's test.




























**Table 4** Total precipitation, mean daily temperature, [a] mean soil Eh, CH$_4$, and N$_2$O

fluxes over the 4 winter fallow seasons.

| Winter fallow season | Precipitation (mm) | Temperature (℃) | Soil Eh (mV) | CH$_4$ flux (mg CH$_4$ m$^{-2}$ h$^{-1}$) | N$_2$O flux (µg N$_2$O-N m$^{-2}$ h$^{-1}$) |
|---|---|---|---|---|---|
| 2010 (December 2, 2010 to April 15, 2011) | 404 | 9.1 | 152 ±11 | 0.02 ±0.01 | 5.01 ±0.26 |
| 2011 (November 3, 2011 to April 19, 2012) | 754 | 10.0 | 102 ±13 | 0.18 ±0.08 | 3.11 ±0.31 |
| 2012 (December 5, 2012 to April 15, 2013) | 574 | 9.7 | 141 ±34 | 0.07 ±0.04 | 8.41 ±0.54 |
| 2013 (November 11, 2013 to April 5, 2014) | 661 | 9.4 | 92 ±12 | 0.08 ±0.03 | 7.06 ±0.38 |

Note: [a] mean soil Eh, CH$_4$, and N$_2$O fluxes were the average of 4 treatments.

































**Table 5** Relative mitigating GWPs of GHGs emissions from paddy fields with various land management practices as compared to traditional managements in winter crop season.

| Type | Traditional management | Suggested practice | GHGs | [a] Mitigation potential (%) | | | | Reference |
|---|---|---|---|---|---|---|---|---|
| | | | | WS | ES | LS | Annual | |
| Double rice | Winter fallow without drainage nor tillage | Drainage | $CH_4$ and $N_2O$ | 30 | 27 | 21 | 24 | This study |
| | | Tillage | $CH_4$ and $N_2O$ | -60 | 17 | 15 | 15 | |
| | | Drainage combined with tillage | $CH_4$ and $N_2O$ | 0 | 35 | 29 | 32 | |
| Single rice | Winter wheat with drainage | Tillage | $CH_4$ and $N_2O$ | 21 | 14 | | 15 | (Zhang et al., 2015) |
| Single rice | Winter ryegrass with drainage | Tillage | $N_2O$ | [b] N.m. | 22 | | N.m. | (Bayer et al., 2015) |
| Single rice | Winter wheat with drainage | Tillage | $CH_4$ and $N_2O$ | 38 | N.m. | | N.m. | (Yao et al., 2013) |
| Single rice | Winter fallow and continuous flooding | Oil-seed rape with drainage and tillage | $CH_4$ and $N_2O$ | 4 | 57 | | 43 | (Zhang et al., 2012) |
| Single rice | Winter fallow without drainage nor tillage | Drainage | $CH_4$ | N.m. | 71 | | >71 | (Shiratori et al., 2007) |
| Single rice | Winter fallow with drainage but non-tillage | tillage | $CH_4$, $N_2O$, and $CO_2$ | -21 | N.m. | | N.m. | (Liang et al., 2007) |
| Single rice | Winter fallow and continuous flooding | Wheat with drainage | $CH_4$ and $N_2O$ | 59 | 55 | | 56 | (Jiang et al., 2006) |
| | | Oil-seed rape with drainage | $CH_4$ and $N_2O$ | 53 | 57 | | 56 | |
| Single rice | Winter fallow and continuous flooding | Wheat with drainage | $CH_4$ | 100 | 30 | | 59 | (Cai et al., 2003) |
| Single rice | Winter fallow and continuous flooding | Wheat with drainage | $CH_4$ | N.m. | 68 | | >68 | (Cai et al., 1998) |

Note: WS, ES, and LS means winter fallow season, early-rice season and late-rice season, respectively; annual is the total of winter and rice seasons; [a] Mitigation potential of combined gases was calculated on the basis of $CO_2$ equivalents by assuming GWPs for $CH_4$ and $N_2O$ as 28 and 265 times the equivalent mass of $CO_2$ over a 100-year period (Myhre, 2013): GWPs ($CH_4$ + $N_2O$ + $CO_2$) = ($CH_4 \times 28$) + ($N_2O \times 265$) + ($CO_2 \times 1$); [b] N.m. indicates no measurements.


















**Table 6** Contents of total C (g kg$^{-1}$) and total N (g kg$^{-1}$) in rice stubble.

| Tillage time | Treatment | Total C | Total N | C/N | Tillage time | Treatment | Total C | Total N | C/N |
|---|---|---|---|---|---|---|---|---|---|
| After late-rice harvest in 2011 | TD | 338 | 6.9 | 49 | After late-rice harvest in 2012 | TD | 368 | 8.7 | 42 |
| | TND | 314 | 7.8 | 40 | | TND | 364 | 7.1 | 51 |
| Before early-rice transplanting in 2012 | NTD | 356 | 12.7 | 28 | Before early-rice transplanting in 2013 | NTD | 404 | 12.8 | 32 |
| | NTND | 374 | 10.4 | 36 | | NTND | 397 | 13.4 | 30 |

































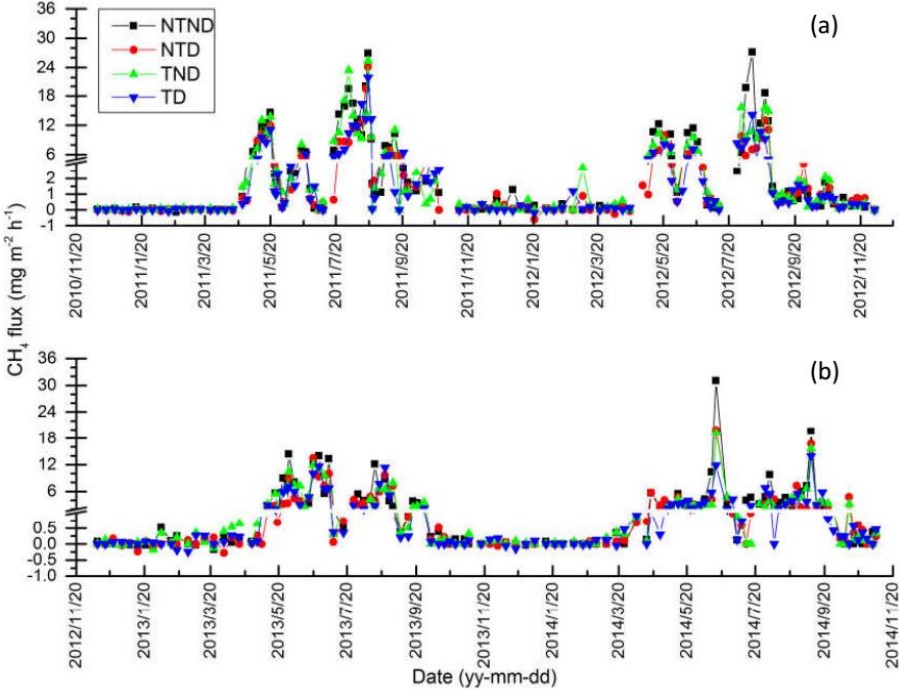


**Figure 1** Seasonal variation of CH₄ emission from 2010 to 2014.




















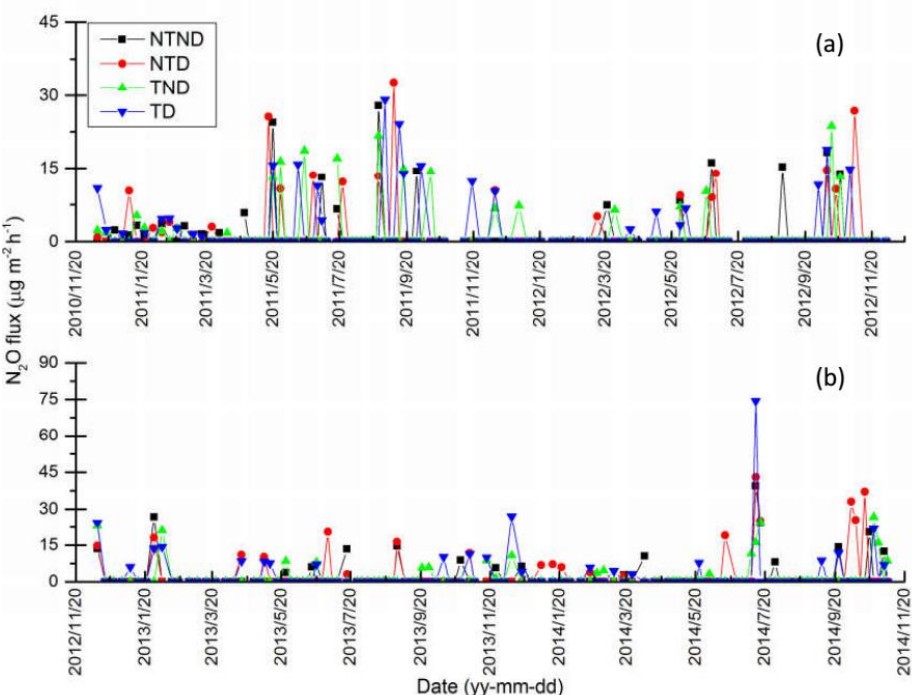

**Figure 2** Seasonal variation of N₂O emission from 2010 to 2014.






















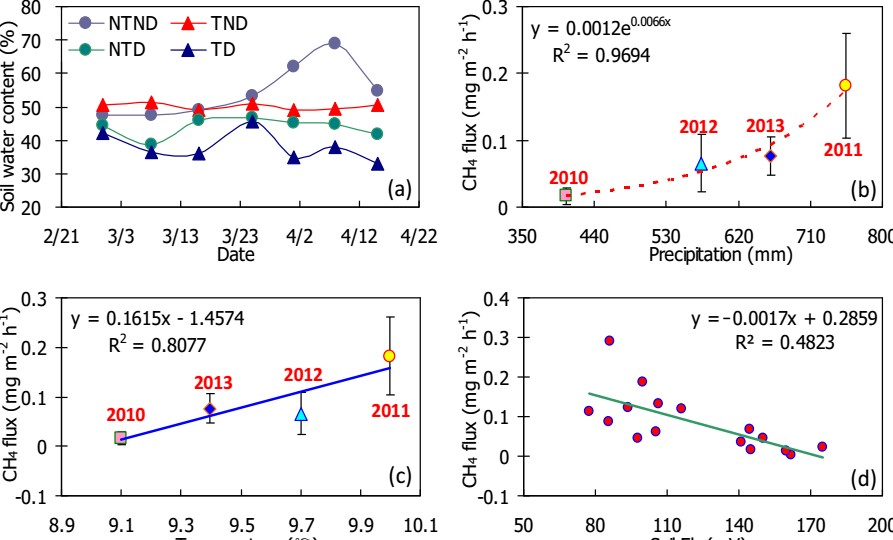

**Figure 3** Soil water content in 2010 winter fallow season (a) and the relationships between mean $CH_4$ emission and total winter precipitation (b), and mean daily air temperature (c) and soil Eh (d) over the 4 winter fallow seasons (Data from Table 4).

























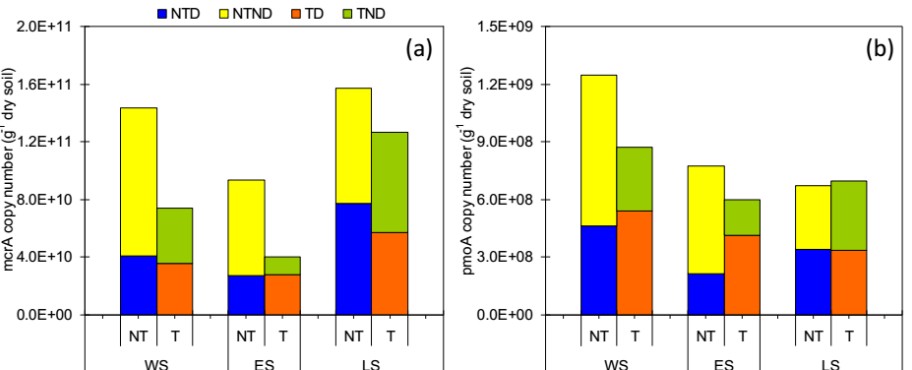

**Figure 4** The abundance of methanogens and methanotrophs populations in paddy soil from 2013 to 2014, WS, ES, and LS means winter fallow season, early-rice season, and late-rice season, respectively.
