# Peer review of "Abstract. Traditional land management (no tillage, no drainage, NTND) during the winter fallow season results in substantial CH4 and N2O emissions from double-rice fields in China. A field experiment was conducted to investigate the effects of drainage and tillage during"

_Atmospheric Chemistry and Physics, 2016_

## Referee Comment (RC1) · Anonymous Referee #1 · 2 Jun 2016

This study investigated CH4 and N2O fluxes from a Chinese double-rice field and the responses to drainage and tillage in winter fallow season for 4 years, estimated the mitigation potential of drainage and tillage, and finally suggested the optimal land management strategies for reducing GWPs of CH4 and N2O emissions in the double rice-cropping systems. More importantly, reasons for decreasing CH4 and N2O emissions were well demonstrated by the measurements of total C and N contents and methanogens. The study provided useful agricultural strategies to mitigate global greenhouse gas emissions from Chinese double-rice fields. The experiment is well designed, and the high-quality data are well presented. The main conclusions are supported by the data. In general, this work is timely and very important with respect to our knowledge of options in winter fallow season for mitigating GWPs in the typical Chinese paddy fields. In particular, the paper made a good analysis of available data

and discussed in detail. The manuscript is well presented, and the English is generally well written, although it has a potential to be improved. Overall, I do not have any major concerns but recommend it to be accepted by Atmospheric Chemistry and Physics.
* * *

---

## Referee Comment (RC2) · Anonymous Referee #2 · 7 Jul 2016

This paper reported 4 years fields experiment results to show how both CH4 and N2O emissions from double-rice paddies affect by drainage and tillage managements in winter fallow season in a typical subtropical climate zoon in Southern China. The global warming potentials (GWPs) from CH4 and N2O, greenhouse gas intensity per yield (GHGI) were also estimated in this paper. The data shown in the paper was reliable and calculation and statistical analyses were suitable. Please consider the minor points shown below for improving this manuscript.

1. The title should be clear, it can be changed as "Options of drainage and tillage managements in winter fallow season for mitigating global warming potential of a double-rice field in China".

2. Line 35, (WMO, 2014) can be renewed to (WMO, 2015). Also (WMO, 2015) should

be listed in References.

3. Line 45, (FAOSTAT, 2013), same as above.

4. Line 47, (Yearbook, 2013), same as above.

5. Line 78, before "In addition", one recent paper (Biol. Fertil. Soils (2016) 52:739–748) can be referred here.

6. Line 103, (Soil Survey Staff, 1975) was not found in References.

7. Line 117-125, the management of rice straw from early rice season was not explained here.

8. Line 146, (Myhre, 2013) as not found in References. It should be (Myhre et al., 2013) or (IPCC, 2013).

9. Line 347, "Parashar et al., 1993" should be before "Cai et al., 2003".

10. Line 467, delete "yr-1" before (Table 3).

11. Line 496, please check the subscript of N2O and CH4 in this manuscript.

12. Line 497, it should be "Biol. Fertil. Soils," not Biol. Fert. Soils.,". The style of with or without DOI number should be consistent, for example, in line 525

13. Table 3, delete "yr-1" from the unit of GHGI.

14. Table 5, (Myhre, 2013) as not found in References. It should be (Myhre et al., 2013) or (IPCC, 2013).

15. Table 6, The C/N ratios of rice stubble increased after winter fallow season was easily understood, but it was confused readers why there were no data for NTD and NTND after late-rice harvest, and for TD and TND before early-rice transplanting.

16. Figure 4, putting NTD and NTND, and TD and TND in same bar graph were not suitable, there are independent treatments.

---

## Author Comment (AC1) · 25 Jul 2016

Thanks for your positive comments and useful suggestions. For more clarity, the language has been improved and perfected, and please refer to the revised manuscript for the detailed revisions.
* * *

---

## Author Comment (AC2) · 25 Jul 2016

Anonymous Referee #2: This paper reported 4 years fields experiment results to show how both CH4 and N2O emissions from double-rice paddies affect by drainage and tillage managements in winter fallow season in a typical subtropical climate zoon in Southern China. The global warming potentials (GWPs) from CH4 and N2O, greenhouse gas intensity per yield (GHGI) were also estimated in this paper. The data shown in the paper was reliable and calculation and statistical analyses were suitable. Please consider the minor points shown below for improving this manuscript.

1. The title should be clear, it can be changed as "Options of drainage and tillage managements in winter fallow season for mitigating global warming potential of a double-rice field in China". It is a good idea, thanks. For more clear and concise however, the

title is supposed to be changed as "Drainage and tillage in winter fallow season mitigate global warming potential of a double-rice field in China" (Line 1∼2 in the revised manuscript).

2. Line 35, (WMO, 2014) can be renewed to (WMO, 2015). Also (WMO, 2015) should be listed in References. Thanks for your suggestions. The data and reference have been updated in the text (Line 34∼35 in the revised manuscript). In addition, the Reference has been changed in the list (Line 623∼624 in the revised manuscript).

3. Line 45, (FAOSTAT, 2013), same as above. Thanks! The data and reference have been changed (Line 44∼45 in the revised manuscript). Also, the Reference is revised in the list (Line 539∼540 in the revised manuscript).

4. Line 47, (Yearbook, 2013), same as above. Thanks so much! The Reference is revised both in the text (Line 47 in the revised manuscript) and the list (Line 636∼637 in the revised manuscript).

5. Line 78, before "In addition", one recent paper (Biol. Fertil. Soils (2016) 52:739–748) can be referred here. Thanks. It is very useful. The reference is cited in the text (Line 78 in the revised manuscript) and it is supplemented in the list (Line 614∼616 in the revised manuscript).

6. Line 103, (Soil Survey Staff, 1975) was not found in References. Sorry for our carelessness, the reference is supplemented (Line 609∼610 in the revised manuscript).

7. Line 117-125, the management of rice straw from early rice season was not explained here. Thanks for your suggestion. A sentence "After early-rice harvest, rice straw and stubble were all moved out of the plots" is supplemented (Line 125∼126 in the revised manuscript) to describe the rice straw management in the early-rice season.

8. Line 146, (Myhre, 2013) as not found in References. It should be (Myhre et al., 2013) or (IPCC, 2013). Sorry for our carelessness. It has been revised in the text (Line 148

in the revised manuscript) and supplemented in the list (Line 585~591 in the revised manuscript).

9. Line 347, "Parashar et al., 1993" should be before "Cai et al., 2003". Thanks. It has been revised (Line 350 in the revised manuscript). Additionally, similar problems in the text are all corrected.

10. Line 467, delete "yr-1" before (Table 3). Sorry for our carelessness. It has been deleted (Line 470 in the revised manuscript). Moreover, similar problems in Table 3 and Abstract are all revised.

11. Line 496, please check the subscript of N2O and CH4 in this manuscript. 12. Line 497, it should be "Biol. Fertil. Soils," not Biol. Fert. Soils.,". The style of with or without DOI number should be consistent, for example, in line 525. Sorry for our carelessness. There (11 and 12) are all changed (Line 499~500 in the revised manuscript). In addition, the DOI numbers in the References are all deleted, please carefully refer to the list, thanks.

13. Table 3, delete "yr-1" from the unit of GHGI. It has been revised (Table 3 in the revised manuscript).

14. Table 5, (Myhre, 2013) as not found in References. It should be (Myhre et al., 2013) or (IPCC, 2013). Thanks. It has been revised (Table 5, Line 815 in the revised manuscript).

15. Table 6, The C/N ratios of rice stubble increased after winter fallow season was easily understood, but it was confused readers why there were no data for NTD and NTND after late-rice harvest, and for TD and TND before early-rice transplanting. Thanks for your valuable suggestion. It should be noted that, firstly, there were two different times of tillage, i.e. tilling the field immediately after late-rice harvest in previous winter fallow season (Treatments TD and TND) and prior to early-rice transplanting during the following rice-growing season (Treatments NTD and NTND). Secondly, the contents of

Total C and Total N in rice stubble were sampled and measured before early-rice trans-
planting. That is to say, rice stubble in Treatments TD and TND were buried under the
soil while in Treatments NTD and NTND rice stubble were exposed to the air through-
out the whole winter fallow season. Thereby, we can estimate the effect of tillage in
winter fallow season on the degradation of rice straw by sampling rice stubble before
early-rice transplanting and measuring the Total C and Total N contents. It is thus clear
that, the phrases "after late-rice harvest" and "before early-rice transplanting" were
just the times of soil tillage, not indicating the times of measurement (or times of data
obtained). In addition, the data in Table 6 were from the measurements of rice stub-
ble sampled before early-rice transplanting (Line 113∼115 in the original manuscript).
Nevertheless, Table 6 and its caption are changed for more understandable (Table 6,
Line 832∼833 in the revised manuscript). Please see below.

16. Figure 4, putting NTD and NTND, and TD and TND in same bar graph were not
suitable, there are independent treatments. Thanks! Certainly, it is more reasonable
for showing the four of them apart, and in deed it was done before. Nevertheless,
the Figure 4 is presented like this, and it is still supposed to be kept in the revised
manuscript if the figure won't result in any misunderstandings. There are at least two
reasons. Firstly, we put the measurements of NTD and NTND, and TD and TND in the
same bar graph here mainly for emphasizing the importance of tillage to the abundance
of methanogens and methanotrophs populations. Because the effect of drainage on
the abundance of methanogens and methanotrophs populations in paddy soil is well
known, however, the effect of tillage, particularly the impact of tillage in winter fallow
season on the abundance of methanogens and methanotrophs populations in paddy
soil during the previous winter fallow and following early- and late-rice seasons are
scarcely reported. Secondly, it is more clear and better comparative by putting Treat-
ment tillage (TD and TND) and Treatment non-tillage (NTD and NTND) together.

Table 6 Measurements of total C (g kg−1) and total N (g kg−1) contents in rice stubble
before early-rice transplanting in 2012 and 2013. Year Treatment Total C Total N C/N

2012 TD 338 6.9 49 TND 314 7.8 40 NTD 356 12.7 28 NTND 374 10.4 36 2013 TD 368 8.7 42 TND 364 7.1 51 NTD 404 12.8 32 NTND 397 13.4 30

Please also note the supplement to this comment: http://www.atmos-chem-phys-discuss.net/acp-2016-227/acp-2016-227-AC2-supplement.pdf

**Supplement:**

Dear prof. Natascha Töpfer,

Thanks for giving us the opportunity to revise the paper "Options for mitigating global warming potential of a double-rice field in China" (**MS No.: acp-2016-227**) after public discussions. In the following we will consecutively address the points given by two anonymous referees (*in italic*) and, if appropriate, will make suggestions how to modify the manuscript.

**Anonymous Referee #1:**

*This study investigated CH4 and N2O fluxes from a Chinese double-rice field and the responses to drainage and tillage in winter fallow season for 4 years, estimated the mitigation potential of drainage and tillage, and finally suggested the optimal land management strategies for reducing GWPs of CH4 and N2O emissions in the double rice-cropping systems. More importantly, reasons for decreasing CH4 and N2O emissions were well demonstrated by the measurements of total C and N contents and methanogens. The study provided useful agricultural strategies to mitigate global greenhouse gas emissions from Chinese double-rice fields. The experiment is well designed, and the high-quality data are well presented. The main conclusions are supported by the data. In general, this work is timely and very important with respect to our knowledge of options in winter fallow season for mitigating GWPs in the typical Chinese paddy fields. In particular, the paper made a good analysis of available data and discussed in detail. The manuscript is well presented, and the English is generally well written, although it has a potential to be improved. Overall, I do not have any major concerns but recommend it to be accepted by Atmospheric Chemistry and Physics.*

Thanks for your positive comments and useful suggestions. For more clarity, the language has been improved and perfected, and please refer to the revised manuscript for the detailed revisions.

**Anonymous Referee #2:**

*This paper reported 4 years fields experiment results to show how both CH4 and N2O emissions from double-rice paddies affect by drainage and tillage managements in winter*

*fallow season in a typical subtropical climate zoon in Southern China. The global warming potentials (GWPs) from CH4 and N2O, greenhouse gas intensity per yield (GHGI) were also estimated in this paper. The data shown in the paper was reliable and calculation and statistical analyses were suitable. Please consider the minor points shown below for improving this manuscript.*

*1. The title should be clear, it can be changed as "Options of drainage and tillage managements in winter fallow season for mitigating global warming potential of a double-rice field in China".*

It is a good idea, thanks. For more clear and concise however, the title is supposed to be changed as "Drainage and tillage in winter fallow season mitigate global warming potential of a double-rice field in China" (Line 1~2 in the revised manuscript).

*2. Line 35, (WMO, 2014) can be renewed to (WMO, 2015). Also (WMO, 2015) should be listed in References.*

Thanks for your suggestions. The data and reference have been updated in the text (Line 34~35 in the revised manuscript). In addition, the Reference has been changed in the list (Line 623~624 in the revised manuscript).

*3. Line 45, (FAOSTAT, 2013), same as above.*

Thanks! The data and reference have been changed (Line 44~45 in the revised manuscript). Also, the Reference is revised in the list (Line 539~540 in the revised manuscript).

*4. Line 47, (Yearbook, 2013), same as above.*

Thanks so much! The Reference is revised both in the text (Line 47 in the revised manuscript) and the list (Line 636~637 in the revised manuscript).

*5. Line 78, before "In addition", one recent paper (Biol. Fertil. Soils (2016) 52:739–748) can be referred here.*

Thanks. It is very useful. The reference is cited in the text (Line 78 in the revised manuscript) and

it is supplemented in the list (Line 614~616 in the revised manuscript).

*6. Line 103, (Soil Survey Staff, 1975) was not found in References.*

Sorry for our carelessness, the reference is supplemented (Line 609~610 in the revised manuscript).

*7. Line 117-125, the management of rice straw from early rice season was not explained here.*

Thanks for your suggestion. A sentence "After early-rice harvest, rice straw and stubble were all moved out of the plots" is supplemented (Line 125~126 in the revised manuscript) to describe the rice straw management in the early-rice season.

*8. Line 146, (Myhre, 2013) as not found in References. It should be (Myhre et al., 2013) or (IPCC, 2013).*

Sorry for our carelessness. It has been revised in the text (Line 148 in the revised manuscript) and supplemented in the list (Line 585~591 in the revised manuscript).

*9. Line 347, "Parashar et al., 1993" should be before "Cai et al., 2003".*

Thanks. It has been revised (Line 350 in the revised manuscript). Additionally, similar problems in the text are all corrected.

*10. Line 467, delete "yr-1" before (Table 3).*

Sorry for our carelessness. It has been deleted (Line 470 in the revised manuscript). Moreover, similar problems in Table 3 and **Abstract** are all revised.

*11. Line 496, please check the subscript of N2O and CH4 in this manuscript.*
*12. Line 497, it should be "Biol. Fertil. Soils," not Biol. Fert. Soils.,". The style of with or without DOI number should be consistent, for example, in line 525.*

Sorry for our carelessness. There (11 and 12) are all changed (Line 499~500 in the revised manuscript). In addition, the DOI numbers in the References are all deleted, please carefully refer to the list, thanks.

*13. Table 3, delete "yr-1" from the unit of GHGI.*

It has been revised (Table 3 in the revised manuscript).

*14. Table 5, (Myhre, 2013) as not found in References. It should be (Myhre et al., 2013) or (IPCC, 2013).*

Thanks. It has been revised (Table 5, Line 815 in the revised manuscript).

*15. Table 6, The C/N ratios of rice stubble increased after winter fallow season was easily understood, but it was confused readers why there were no data for NTD and NTND after late-rice harvest, and for TD and TND before early-rice transplanting.*

Thanks for your valuable suggestion. It should be noted that, firstly, there were two different times of tillage, i.e. tilling the field immediately after late-rice harvest in previous winter fallow season (Treatments TD and TND) and prior to early-rice transplanting during the following rice-growing season (Treatments NTD and NTND). Secondly, the contents of Total C and Total N in rice stubble were sampled and measured before early-rice transplanting. That is to say, rice stubble in Treatments TD and TND were buried under the soil while in Treatments NTD and NTND rice stubble were exposed to the air throughout the whole winter fallow season. Thereby, we can estimate the effect of tillage in winter fallow season on the degradation of rice straw by sampling rice stubble before early-rice transplanting and measuring the Total C and Total N contents. It is thus clear that, the phrases "after late-rice harvest" and "before early-rice transplanting" were just the times of soil tillage, not indicating the times of measurement (or times of data obtained). In addition, the data in Table 6 were from the measurements of rice stubble sampled before early-rice transplanting (Line 113~115 in the original manuscript). **Nevertheless**, Table 6 and its caption are changed for more understandable (Table 6, Line 832~833 in the revised manuscript). Please see below.

*16. Figure 4, putting NTD and NTND, and TD and TND in same bar graph were not suitable, there are independent treatments.*

Thanks! Certainly, it is more reasonable for showing the four of them apart, and in deed it was done before. Nevertheless, the **Figure 4** is presented like this, and it is still supposed to be kept in the

revised manuscript if the figure won't result in any misunderstandings. There are at least two reasons. Firstly, we put the measurements of NTD and NTND, and TD and TND in the same bar graph here mainly for emphasizing the importance of tillage to the abundance of methanogens and methanotrophs populations. Because the effect of drainage on the abundance of methanogens and methanotrophs populations in paddy soil is well known, however, the effect of tillage, particularly the impact of tillage in winter fallow season on the abundance of methanogens and methanotrophs populations in paddy soil during the previous winter fallow and following early- and late-rice seasons are scarcely reported. Secondly, it is more clear and better comparative by putting Treatment tillage (TD and TND) and Treatment non-tillage (NTD and NTND) together.

**Table 6** Measurements of total C (g kg$^{-1}$) and total N (g kg$^{-1}$) contents in rice stubble before early-rice transplanting in 2012 and 2013.

| Year | Treatment | Total C | Total N | C/N |
|------|-----------|---------|---------|-----|
| 2012 | TD        | 338     | 6.9     | 49  |
|      | TND       | 314     | 7.8     | 40  |
|      | NTD       | 356     | 12.7    | 28  |
|      | NTND      | 374     | 10.4    | 36  |
| 2013 | TD        | 368     | 8.7     | 42  |
|      | TND       | 364     | 7.1     | 51  |
|      | NTD       | 404     | 12.8    | 32  |
|      | NTND      | 397     | 13.4    | 30  |

Thanks again! If the current manuscript still need revising, please feel free to let me know.

With best regards,

Guang-bin Zhang

State Key Laboratory of Soil and Sustainable Agriculture

Institute of Soil Science, Chinese Academy of Sciences

Nanjing 210008, China

Phone: 0086-25-86881132;

Fax: 0086-25-86881028;

Email: gbzhang@issas.ac.cn